

# Expression analysis of genes related to cold tolerance in *Dendroctonus valens*

Dongfang Zhao, Chunchun Zheng, Fengming Shi, Yabei Xu, Shixiang Zong and Jing Tao

Key Laboratory of Beijing for the Control of Forest Pests, Beijing Forestry University, Beijing, China

## ABSTRACT

Pine beetles are well known in North America for their widespread devastation of pine forests. However, *Dendroctonus valens* LeConte is an important invasive forest pest in China also. Adults and larvae of this bark beetle mainly winter at the trunks and roots of *Pinus tabuliformis* and *Pinus sylvestris*; larvae, in particular, result in pine weakness or even death. Since the species was introduced from the United States to Shanxi in 1998, its distribution has spread northward. In 2017, it invaded a large area at the junction of Liaoning, Inner Mongolia and Hebei provinces, showing strong cold tolerance. To identify genes relevant to cold tolerance and the process of overwintering, we sequenced the transcriptomes of wintering and non-wintering adult and larval *D. valens* using the Illumina HiSeq platform. Differential expression analysis methods for other non-model organisms were used to compare transcript abundances in adults and larvae at two time periods, followed by the identification of functions and metabolic pathways related to genes associated with cold tolerance. We detected 4,387 and 6,091 differentially expressed genes (DEGs) between sampling dates in larvae and adults, respectively, and 1,140 common DEGs, including genes encoding protein phosphatase, very long-chain fatty acids protein, cytochrome P450, and putative leucine-rich repeat-containing proteins. In a Gene Ontology (GO) enrichment analysis, 1,140 genes were assigned to 44 terms, with significant enrichment for cellulase activity, hydrolase activity, and carbohydrate metabolism. Kyoto Encyclopedia of Genes and Genomes (KEGG) classification and enrichment analyses showed that the lysosomal and purine metabolism pathways involved the most DEGs, the highly enriched terms included autophagy—animal, pentose and glucuronate interconversions and lysosomal processes. We identified 140 candidate genes associated with cold tolerance, including genes with established roles in this trait (e.g., genes encoding trehalose transporter, fructose-1,6-bisphosphatase, and trehalase). Our comparative transcriptome analysis of adult and larval *D. valens* in different conditions provides basic data for the discovery of key genes and molecular mechanisms underlying cold tolerance.

# INTRODUCTION

*Dendroctonus valens*, the red turpentine beetle, is a species of bark beetle that mainly attacks the base of the trunk *P. tabuliformis*. Adults generally lay eggs in the phloem at the base of the trunk or 1.5 m below the base. After hatching, larvae consume decaying phloem and

Corresponding author
Jing Tao, taojing1029@hotmail.com

form a common tunnel. Adults and larvae eat the phloem, destroy the cambium, and cut off nutrient transport in swarms, thereby affecting tree growth or even causing death. This damage reduces the economic and landscape value of the tree (*Yan et al., 2005*).

*Dendroctonus valens* was introduced to Shanxi Province in 1998 and spread rapidly due to the abundant *Pinus* hosts and warm and dry climate (*Sun et al., 2013*). The species was introduced to Hebei and Henan in 1999 (*Sun et al., 2004*), Shaanxi and Qinghai in 2001, and Beijing in 2005, and its distribution continued to expand northward. By 2017, it reached to Chaoyang of Liaoning and Chifeng of Inner Mongolia at approximately 41.5°N latitude.

Insect cold tolerance has been studied since the 1960s (*Belehradek, 1957*; *Salt, 1961*). Research in this area has progressed rapidly since the 1990s, in large part owing to theoretical advances related to insect cold tolerance (*Huey et al., 1992*; *Bale, 2002*). Technological and scientific developments have enabled a deeper understanding of cryobiology. Various omics technologies have been used to characterize the molecular mechanisms underlying cold tolerance. Recent studies of cold tolerance in insects have focused on the determination of the supercooling point, survival in low-temperature conditions, the cold tolerance index, and the influence of cold acclimation on insect biology. Transcriptome techniques, such as gene chip technology, expressed sequence tags, serial analysis of gene expression, and RNA sequencing, have been used to identify highly expressed cold-related genes in insects (*Barth et al., 2018*; *Chen et al., 2019*; *Enriquez & Colinet, 2019*). To screen cold-tolerant genes, a variety of tools for studying differential gene expression are available, including cDNA library screening, suppression subtractive hybridization, and microarray screening (*Parker et al., 2018*; *Martinson et al., 2019*; *Wang et al., 2019*). Two-dimensional electrophoresis, mass spectrometry, nuclear magnetic resonance, and affinity chromatography technology have been used to separate and purify antifreeze proteins extracted from insects, analyze the amino acid composition, and explore the structural characteristics and mechanism of action (*Tomalty et al., 2019*; *Xiang et al., 2020*; *Graham et al., 2020*). Extensive research has confirmed that cold acclimation influences cold tolerance in insects, as determined by a variety of physiological and biochemical methods, such as analyses of the difference between dry and wet weights, differential scanning calorimetry, spectrophotometry, and capillary gas spectroscopy (*Holmstrup, 2018*; *Verheyen et al., 2018*; *Pei et al., 2020*).

In view of the expansion of the distribution of *D. valens*, extensive research has focused on biological characteristics (*Liu, Xu & Sun, 2014*; *Zhang et al., 2010*; *Lindeman & Yack, 2015*; *Liu et al., 2017*), risk analyses (*Liu, Xu & Sun, 2014*), biological control (*Storey, Storey & Zhang, 2011*; *Yang, Wang & Zhang, 2014*), symbiotic bacteria (*Wang et al., 2012*; *Raffa, Hanshew & Mason, 2016*; *Wang et al., 2017*), and pheromones (*Chen et al., 2010*; *Xu, Liu & Sun, 2014a*; *Xu, Liu & Sun, 2014b*; *Kelsey & Westlind, 2018*), but studies of the molecular mechanisms underlying cold tolerance in *D. valens* are lacking. Other researches on the cold tolerance of bark beetles are mainly reflected in the physiological and biochemical analysis. Many scholars have measured the low temperature survival, supercooling ability, hemolymph osmotic pressure, lipid and carbohydrate reserves, and concentration of potential cryoprotectant glycerol of other bark beetles (such as *Ips typographus, Pityogenes chalcographus, Dendroctonus armandi, Dendroctonus ponderosae, Pityophthorus juglandis,*

tiger beetles), the seasonal development of physiological indices underlying gradual acquisition of relatively high cold tolerance to determine the possible overwintering strategies were described (*Régnière & Bentz, 2007*; *Koštál et al., 2011a*; *Koštál et al., 2011b*; *Koštál et al., 2014*; *Wang et al., 2016*; *Bleiker, Smith & Humble, 2017*; *Hefty et al., 2017*; *Dai et al., 2019*; *Burns et al., 2020*). In addition, previous work has shown that glycerin enhances the cold tolerance of *D. ponderosae* during overwintering (*Bonnett et al., 2012*; *Fraser et al., 2017*). Cold tolerance is an important indicator of the population density, diffusion, and distribution of insects and can explain adaptation to low temperatures in northern regions. Genes related to cold tolerance in insects have been identified in cold-sensitive *Drosophila melanogaster* (*Hoffmann, 2010*). A number of candidate genes and proteins related to low-temperature responses (e.g., factors involved in metabolism, the epidermis, cytoskeleton, immune function, and signal transduction (*Vesala et al., 2012*; *Isobe, Takahashi & Tamura, 2013*; *Parker et al., 2015*)) have been reported in drosophila, including heat shock proteins (HSPs) (*Colinet et al., 2013*; *Enriquez & Colinet, 2019*) and Frost genes (*Colinet et al., 2013*; *MacMillan et al., 2017*).

Transcriptomics, as a bridge linking genomics and proteomics, has been applied in many fields of biology. It is a powerful tool to evaluate gene expression differences, phylogenetic relationships, rates of protein evolution, genotype–phenotype correlations, chromosome structure, and regulatory mechanisms, as well as for the development of molecular markers and functional genomics research (*Oppenheim et al., 2015*). In recent years, with the reduction of sequencing costs, transcriptome sequencing has become increasingly common, particularly RNA-Seq analyses using high-throughput sequencing technology. An obvious advantage of RNA-Seq is the ability to directly assess sequences without prior knowledge of gene structure and to identify novel transcripts. This method enables studies of gene expression in non-model insects and species without sequenced genomes (*Malone & Oliver, 2011*; *Alvarez, Schrey & Richards, 2015*; *Jazayeri, Munoz & Romero, 2015*).

In this study, high-throughput transcriptome sequencing was used to explore key genes and metabolic pathways related to responses to low temperatures in *D. valens*. These results provide a foundation for further molecular and functional studies of this trait and provide a theoretical reference for the development of a prevention and control strategy.

## MATERIALS AND METHODS

### Experimental insects

*Dendroctonus valens* late instar larvae and adults were collected from the field on January 19, 2019, and May 11, 2019, in Heilihe National Nature Reserve, Ningcheng County, Chifeng City, Inner Mongolia (Longitude: 118.43°E; Latitude: 41.41°N; Altitude: 1,050 m). Liu Yushan, head of Chifeng Forest Protection Station in Inner Mongolia, has approved sample collection. The average temperature was −7.36 °C and 18.82 °C on these months respectively (http://data.cma.cn/). The samples were stored in liquid nitrogen, brought to the laboratory, and stored at −80 °C until RNA extraction.

## RNA extraction, cDNA library construction and sequencing

Total RNA was extracted from *D. valens* collected in January and May using TRIzol reagent (Ambion, Austin, TX, USA) and the RNeasy Plus Mini Kit (No. 74,134; Qiagen, Hilden, Germany) following the manufacturers' instructions. The larvae under cold and normal temperature are referred to as CL and NL respectively, and the adults under cold and normal temperature are referred to as CA and NA respectively. The biological material was whole insect. Each sample consisted of a single individual, and three biological replicates were evaluated per sample. The purity, concentration, and integrity of the total RNA samples were measured using the NanoDrop2000 (IMPLEN, Westlake Village, CA, USA), Agilent 2100 (Agilent Technologies, Santa Clara, CA, USA) and 1.0% agarose gel electrophoresis. Library construction required a sample of $\geq 2\,\mu g$, a concentration of $\geq 50$ ng/$\mu$l, clear RNA bands, no contamination by impurities (such as pigments, proteins, and sugars), a ratio of 28/23S: 18/16S of $>1$, RIN value of $\geq 6.5$, $OD_{260/280} \geq 1.8$, and $OD_{260/230} \geq 1.5$.

RNA purification, library construction and sequencing were performed at Shanghai Majorbio Bio-pharm Biotechnology Co., Ltd. (Shanghai, China) according to the manufacturer's instructions (Illumina, San Diego, CA). The RNA-seq transcriptome libraries were prepared using Illumina TruSeqTM RNA sample preparation Kit (San Diego, CA). Poly(A) mRNA was purified from total RNA using oligo-dT-attached magnetic beads and then fragmented by fragmentation buffer. Taking these short fragments as templates, double-stranded cDNA was synthesized using random hexamer primers. Then the synthesized cDNA was subjected to end-repair, phosphorylation and 'A' base addition according to Illumina's library construction protocol. Libraries were size selected for cDNA target fragments of 200–300 bp on 2% low range ultra agarose followed by PCR amplified using Phusion DNA polymerase (New England Biolabs, Boston, MA) for 15 PCR cycles. Subsequently, the library preparations were sequenced on an Illumina Hiseq xten/NovaSeq 6000 sequencer (Illumina, San Diego, CA) for $2 \times 150$ bp paired-end reads. All raw sequence data were deposited in the National Center for Biotechnology Information (NCBI) Sequence Read Archive (SRA) database under BioProject accession number PRJNA609406. The assembled sequences have been submitted to the Gene Expression Omnibus (GEO) with accession number GSE156139. This Transcriptome Shotgun Assembly project has been deposited at GenBank under the accession GIWV00000000. The version described in this paper is the first version, GIWV01000000.

## Sequence data analysis and assembly

The Illumina platform converts sequenced image signals into text signals through CASAVA Base Calling and stores these raw data in fastq format. Data for each sample were identified according to the index sequence for subsequent analyses. Fastx_toolkit_0.0.14 (http://hannonlab.cshl.edu/fastx_toolkit/) was used to analyze the base quality, $A/T/G/C$ base content, and base error rate distribution for each sample, as well as the Q20, Q30, and GC contents. SeqPrep (https://github.com/jstjohn/SeqPrep) and Sickle (https://github.com/najoshi/sickle) were used to eliminate raw sequencing data, including adapter sequences, at 3′ end quality reads of $<20$, sequences with a high N rate of $>10\%$ (where N indicates uncertain base information), and sequences of $<30$ bp after quality

trimming. All sequenced reads after quality control were performed de novo assembly using Trinity_v2.8.5 (https://github.com/trinityrnaseq/trinityrnaseq/wiki). TransRate _v1.0.3 (http://hibberdlab.com/transrate/) was used to filter and optimize the sequences obtained from the de novo assembly of the transcriptome. By the evaluation of common errors, such as chimeras, structural errors, base errors, and incomplete assemblies, a quality score for each contig was obtained, and these were integrated to obtain an overall score for filtering optimization (*Smith-Unna et al., 2016*). CD-HIT_v4.5.7 (threshold: 0.99) (http://weizhongli-lab.org/cd-hit/) was used to remove redundant and similar sequences by a sequence clustering method (*Li & Godzik, 2006*). The completeness of the assembly was assessed with the Benchmarking Universal Single-Copy Orthologs (BUSCO, version 3.0.2, http://busco.ezlab.org) tool using the metazoa_odb9 (2016-02-13) dataset. BUSCO can identify complete, duplicated, fragmented and missing genes, analyze the completeness of annotated genes and transcriptomes, and perform similar quality comparisons of different data sets (*Simao et al., 2015*; *Waterhouse et al., 2018*).

## Functional annotation of genes

BLAST was used to obtain homologous sequences by searching against several public protein databases, including the NCBI non-redundant (Nr) protein database, the Swiss-Prot protein database, the Protein Family (Pfam) database, the Cluster of Orthologous Groups (COG) database, the GO database (*Camacho et al., 2009*), and the KEGG database (*Kanehisa, 2000*). DIAMOND_v0.8.37.99 (E-value: 1e−5) was used for comparisons with the NR library, the annotation of *D. valens* transcript sequences from homologous sequences in other species, and functional annotations of proteins (*Buchfink, Xie & Huson, 2015*). BLAST2GO_2.5.0 (E-value: 1e−5) was used to evaluate functions of genes and gene products according to the three major GO categories, biological processes, cellular components, and molecular functions (*Conesa et al., 2005*). KOBAS_2.1.1 (E-value: 1e−5) was used for a KEGG pathway analysis and DIAMOND_v0.8.37.99 (E-value: 1e−5) was used to identify COG groups (*Xie et al., 2011*).

## Differential gene expression analysis

Bowtie2_2.3.5.1 (https://sourceforge.net/projects/bowtie-bio/files/bowtie2/2.3.5.1/) was used to compare the sequence data after quality control with assembled non-redundant transcriptome sequences. The read counts of a gene in each sample were estimated by mapping clean reads to the Trinity transcripts assembled by RSEM_v1.2.31 (*Li & Dewey, 2011*). The Fragments Per Kilobase per Million mapped reads (FPKM) method can be used for standardization to eliminate the effect of gene length and sequencing volume on estimates of expression levels. DESeq2 (Version 1.24.0) and the negative binomial distribution (http://bioconductor.org/packages/stats/bioc/DESeq2/) were used for a statistical analysis to obtain the differentially expressed gene set between wintering periods in *D. valens*. Differentially expressed genes can be divided into up-regulated and down-regulated genes. The samples collected in May are used as controls, the expression level of a gene in January is stronger than that in May, which is an up-regulated gene, and vice versa. In the differential expression analysis, the Benjamini–Hochberg method was

used to correct for multiple testing; the corrected *p*-value is referred to as *p*-adjust, and the default value for *p*-adjust was used as the threshold for screening differentially expressed genes (*Benjamini & Yekutieli, 2001*). In particular, the default screening parameters were *p*-adjust < 0.01 and |log2FC|≥ 1. FC (fold change) for the identification of a difference in the expression of unigenes between the two groups of samples. The screening results are presented in the form of a volcano plot.

To identify functions associated with the differentially expressed gene set, Goatools_0.6.5 was used to perform a GO enrichment analysis of the genes using Fisher's exact test (*Klopfenstein et al., 2018*). The Bonferroni method was used to correct *p*-values. A corrected *p*-value (FDR) of <0.05 indicated significant enrichment for the GO term. To systematically analyze the metabolic pathways involving the differentially expressed genes and the functions of their gene products, Fisher's exact tests were used. Similar to the GO enrichment analysis, when the *p*-value (FDR) was less than 0.05, the KEGG pathway was considered significantly enriched in the gene set (*Du et al., 2014*; *Kanehisa et al., 2017*).

## Gene expression validation by real-time quantitative PCR

To validate the reliability of RNA-Seq data, 14 common DEGs both larvae and adults were selected for an quantitative real-time PCR (qRT-PCR) assay, using TUB and PRS18 as reference genes for larvae and adults, respectively (*Zheng et al., 2020*). Primers were designed using Primer 3Plus (*Untergasser et al., 2007*) and synthesized by Ruibo Xingke Biotechnology Co., Ltd. (Beijing, China) (Table S1). A 5 × gradient dilution of the cDNA of adults and mature larvae was used as a template to draw the standard curve to determine the amplification efficiency of primers. The target fragment was amplified by conventional PCR and the optimal reaction conditions were explored. The SYBR Green dye method was used for qPCR, and the procedure was performed according to the instructions provided with the Fluorescence Quantitative Reaction Kit (Roche, Basel, Switzerland). The reaction system (12.5 µL) included SYBR® Premix Ex TaqII (6.25 µL), 0.5 µL of 10 µmol/L forward primer, 0.5 µL of 10 µmol/L reverse primer, cDNA (1 µL), and ddH$_2$O (4.25 µL), mixed well on ice. PCR conditions were as follows: 95 °C for 3 min, 40 cycles at 95 °C for 10 s, 60 °C for 30 s, and 65 °C to 95 °C in increments of 0.5 °C for 5 s to generate melting curves. Each reaction was performed for three biological and three technical replicates. The experimental data were normalized by the $2^{-\Delta\Delta t}$ method (*Livak & Schmittgen, 2001*). GraphPad Prism 7 was used for statistical analyses and generating plots of real-time PCR results.

## RESULTS

### Sequencing and assembly of *D. valens* transcriptomes

The Illumina sequencing platform was used to sequence 12 samples of *D. valens*. Over 6.1 Gb of clean data were obtained for each sample, with Q30 ≥ 93.21%, Q20 ≥ 97.82%, and a GC content of 42.98% (Table 1). Trinity was used to assemble all clean reads from scratch, and 90,404 transcripts were obtained. After clustering and reducing redundancy, 50,677 unigenes were obtained. The average transcript length after assembly was 911.08 bp, N50 was 1,803 bp, and the BUSCO score was C:89.2%[S:84.9%,D:4.3%], F:6.9%, M:3.9%,

**Table 1  Statistical summary of sequencing data for 12 cDNA samples from *D. valens*.**

| Sample | Raw bases | Clean bases | Error rate (%) | Q20 (%) | Q30 (%) | GC content (%) |
|--------|-----------|-------------|----------------|---------|---------|----------------|
| CL_1 | 7625015592 | 7435581614 | 0.0258 | 97.81 | 93.21 | 42.31 |
| CL_2 | 6606424254 | 6408262061 | 0.0258 | 97.82 | 93.23 | 42.14 |
| CL_3 | 7213969734 | 7003588404 | 0.0253 | 98.01 | 93.71 | 43.54 |
| NL_1 | 6855725858 | 6669592971 | 0.0247 | 98.16 | 94.46 | 49.37 |
| NL_2 | 6275942634 | 6141265236 | 0.0245 | 98.22 | 94.58 | 46.91 |
| NL_3 | 6658060516 | 6433890287 | 0.0251 | 97.96 | 94.05 | 50.28 |
| CA_1 | 7973857302 | 7780530188 | 0.0235 | 98.61 | 95.64 | 42.34 |
| CA_2 | 7687548920 | 7512555521 | 0.0237 | 98.55 | 95.48 | 42.12 |
| CA_3 | 7478884134 | 7284498797 | 0.0234 | 98.67 | 95.72 | 39.96 |
| NA_1 | 8155209510 | 7979742248 | 0.0235 | 98.63 | 95.62 | 38.6 |
| NA_2 | 8403092318 | 8234779825 | 0.0233 | 98.71 | 95.86 | 40.07 |
| NA_3 | 7578902910 | 7436520894 | 0.0236 | 98.62 | 95.59 | 38.13 |

Notes.

In sample names, CL indicates larvae collected in January, NL indicates larvae collected in May, CA indicates adults collected in January, and NA indicates adults collected in May.

n:978 (Table S2). Based on these parameters, the data quality and reliability were high, meeting the requirements for further analyses.

## Functional annotation of genes

The assembled sequences were compared with sequence data in the Nr, Pfam, COG, Swiss-Prot, KEGG, and GO databases using BLAST (e $\leq$ 10-5). Functional annotations were obtained for 28,218 sequences, accounting for approximately 47.86% of all unigene sequences in the transcriptome (Fig. 1). Among these, 27,047 unigenes were annotated in the Nr database, accounting for the highest proportion (45.88%), followed by Pfam (33.84%), GO (32.22%), Swiss-Prot (33.32%), and KEGG (24.94%). The *E*-value distribution, identity distribution, and species distribution were used to further analyze homology between Illumina sequences and those in the Nr database. Some sequences in the samples were contaminated by nematodes, we deleted the sequences with nematode contamination and added annotations again. Based on the *E*-value distribution, 68.99% of the annotated unigenes (12,284) had very high homology with proteins in the Nr database (*E*-value <1e−30), and the other sequences had matches with E-values ranging from 1e−30 to 1e−5 (Fig. 2A). Further analysis showed that 93.33% of the sequences had similarities of >60% to those in the Nr database (Fig. 2B). With respect to species, the annotated sequences had the highest degree of match to those of *D. ponderosae*, with 12,772 sequences and a matching rate of 71.73% (Fig. 2C).

## Differentially expressed gene screening results of *D. valens* under different temperature conditions

The results showed that 4,387 DEGs were found among the larvae sampled in January and May. Compared with larvae collected in May, 24,85 (56.64%) of genes collected in January were up-regulated, 1,902 (43.36%) were down-regulated (Fig. 3A). There were
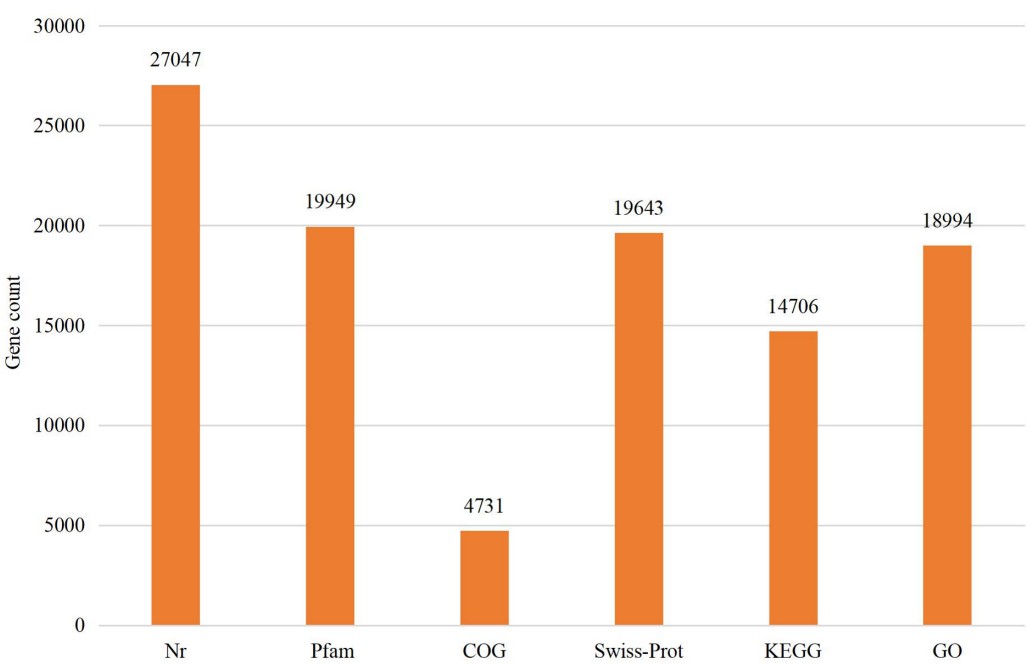

**Figure 1   Statistical summary of the functional annotations of unigenes in public databases.**

6,091 DEGs in adults, including 3,670 (60.25%) up-regulated genes and 2,421 (39.75%) down-regulated genes in January compared with May (Fig. 3B).

As shown in a Venn diagram in Fig. 4, 1,140 genes were differentially expressed between January and May in both larvae and adults (Table S3), 4,951 genes were differentially expressed only in adults, and 3,247 genes were differentially expressed only in larvae. Among the common DEGs in adults and larvae, 218 lacked functional annotations in any database, 505 were down-regulated in adults and larvae, and 370 were up-regulated. In addition, 168 DEGs were up-regulated in larvae and down-regulated in adults, only 97 DEGs were down-regulated in larvae and up-regulated in adults. We speculated that the common DEGs in adults and larvae under different field temperatures might play an important role in the response to low temperatures; accordingly, we focused on these genes in further functional analyses.

## Classification of common DEGs

The 1,140 common DEGs were assigned to three GO domains: biological process (BP), cellular component (CC), and molecular function (MF), involving 44 total GO terms (Fig. 5). In the BP category, the most frequent terms were cellular process (348 genes), followed by metabolic process (275 genes). In the CC category, the most frequent terms were cell part and membrane part (371 and 250, respectively). In the MF category, with binding accounting for the largest proportion of genes (408 genes). A KEGG analysis revealed that 1,421 unigenes belonged to 287 pathways (Table 2). There were 19 metabolic pathways containing 10–20 genes, and the remaining pathways all had fewer than 10 genes.

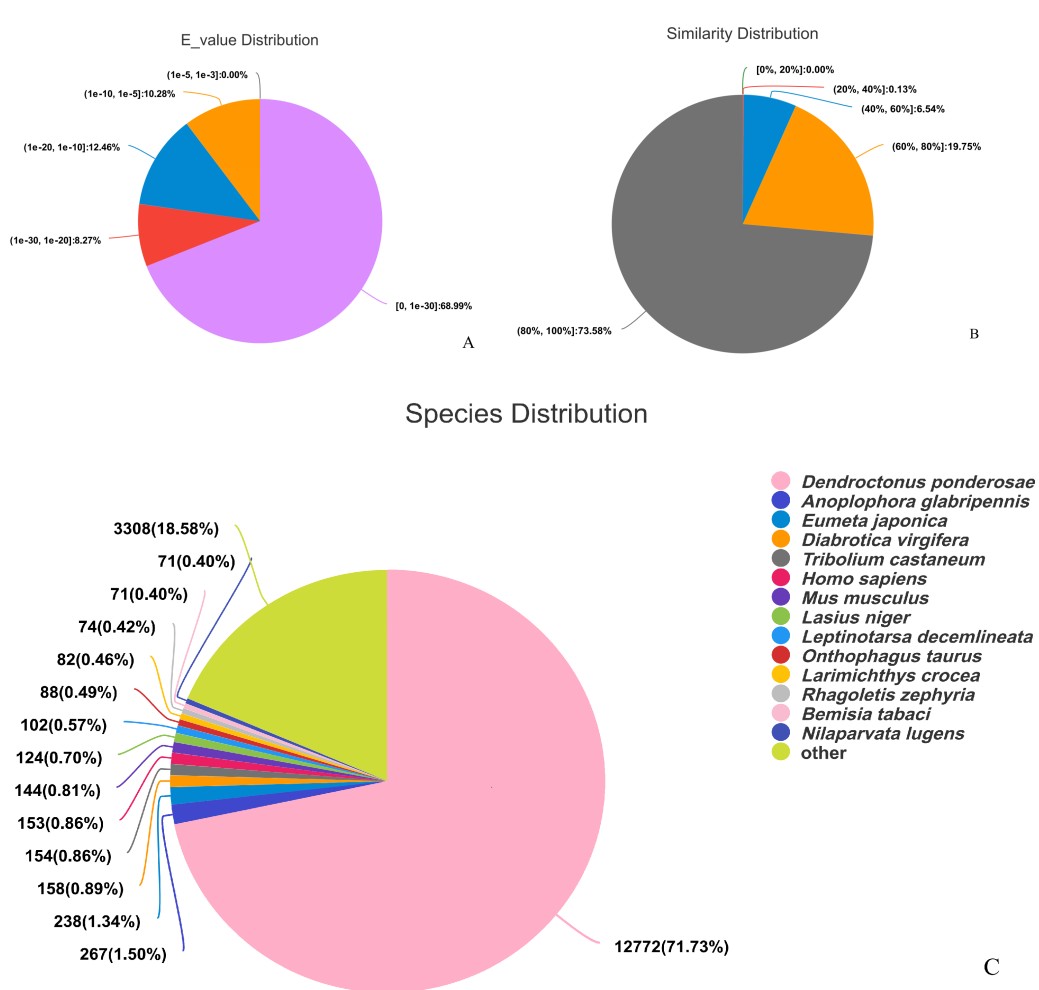

**Figure 2** Pie charts showing distributions of BLAST matches for the *D. valens* unigenes with respect to (A) e-values (B) similarity, and (C) species.

The pathways identified in this analysis were mainly involved in the processes of translation, endocrine system, transport catabolism, carbohydrate metabolism, signal transduction, nucleotide metabolism, and cell growth and death, indicating that signal transduction and substance transport activities may be important in *D. valens*.

## Enrichment analysis of common DEGs

In an evaluation of the top 25 common DEGs (Table 3), 12 terms in BP and 13 terms in MF were significantly enriched. The entry carbohydrate metabolic process (GO: 0005975) was significantly enriched in BP, indicating that carbohydrate metabolism in *D. valens* might contribute to overwintering. In the MF category, the two most significantly enriched GO terms were hydrolase activity, acting on glycosyl bonds (GO: 0016798) and hydrolase activity, hydrolyzing O-glycosyl compounds (GO: 0004553), further illustrating that the molecular activity of *D. valens* is strong under low temperatures. Among the common up-regulated DEGs between sampling dates in adults and larvae, the top two were

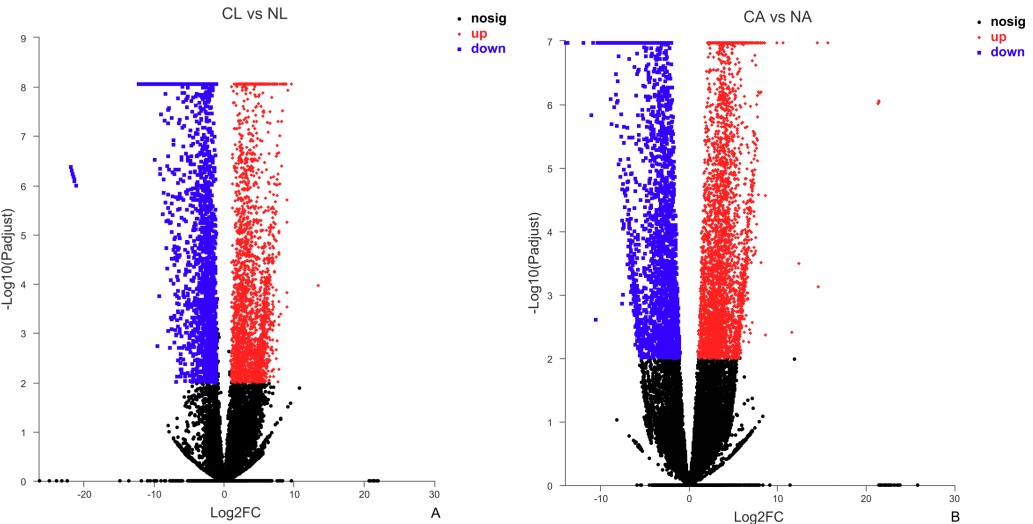

**Figure 3** **Distribution of DEGs between January and May in different sample types.** (A) Distribution of DEGs in larvae. (B) Distribution of DEGs in adults. Volcano plot displays the relationship between the fold change and P-adjusted. Each point in the figure represents a specific unigene. Red dots represent significantly up-regulated unigenes, blue dots represent significantly down-regulated unigenes, and black dots represent unigenes that are not differentially expressed.

nucleic acid binding (GO:0003676), RNA metabolic process (GO:0016070) (Fig. S1A). Among the common down-regulated DEGs, carbohydrate metabolic process (GO: 0005975), hydrolase activity, acting on glycosyl bonds (GO: 0016798), hydrolase activity, hydrolyzing O-glycosyl compounds (GO: 0004553), cellulase activity (GO: 0008810), catalytic activity(GO:0003824) and other terms were significantly enriched (Fig. S1B).

A KEGG analysis revealed that common DEGs could be classified into 287 pathways, the highly enriched terms included pentose and glucuronate interconversions (map00040), lysosome (map04142), and autophagy-animal (map04140), which belonged to carbohydrate metabolism and transport and catabolic metabolism pathways, respectively. It is possible that these processes contribute to overwintering in *D. valens.* Among the common up-regulated DEGs between sampling dates in adults and larvae, we detected significant enrichment for homologous recombination (map03440) and basal transcription factors (map03022) (Fig. S2A). The enrichment results for common down-regulated DEGs in adults and larvae were similar to common DEGs for adults and larvae, including lysosome (map04142), pentose and glucuronate interconversions (map00040), galactose metabolism (map00052), and other terms (Fig. S2B).

## Screening of genes related to cold tolerance among common DEGs

Based on previous studies of insect cold tolerance and the keywords obtained in manual searches of the annotation information for the common DEG set, we selected 140 candidate genes related to cold tolerance. In total, 72 genes encoded enzymes and proteins related to the synthesis and metabolism of cryoprotectants, including low-molecular-weight polyols, trehalose, and low-molecular-weight amino acids (Fig. 6).

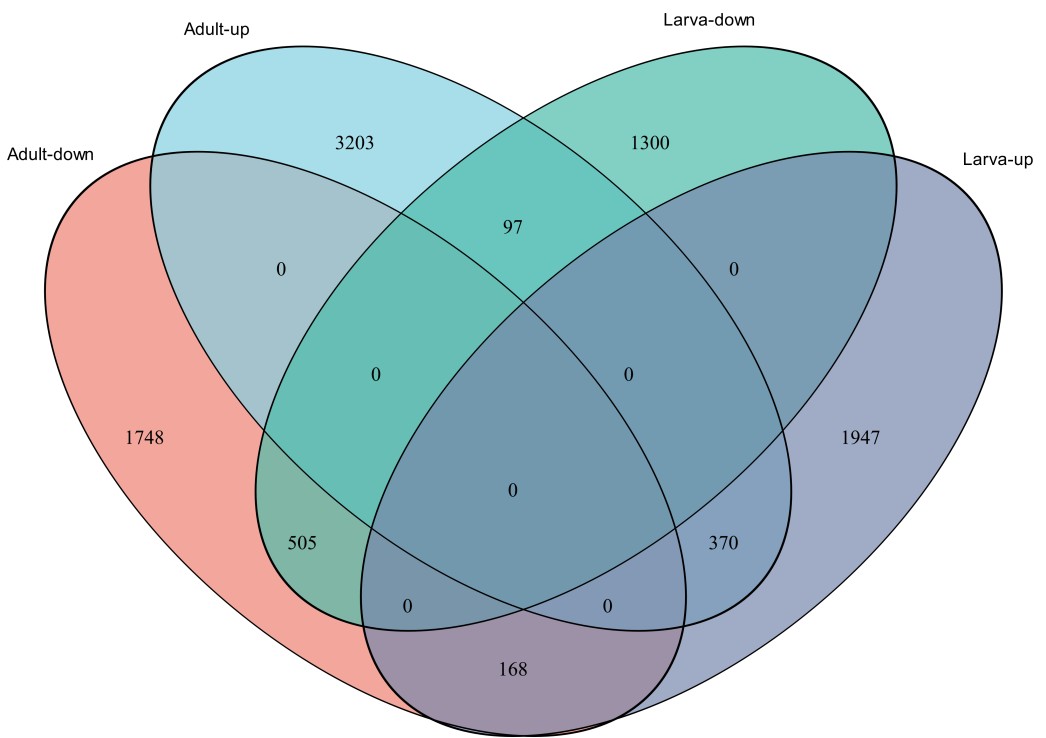

**Figure 4** Venn diagram showing the number of commonly up- and down-regulated genes between the two different stages in *D. valens*.

In addition, two genes encoding the cytoprotective protein *HSP*, 12 genes encoding cytoskeletal proteins, nine genes involved in repair, detoxicant, antioxidant defense, 10 genes related to signal transduction, 1 gene related to cell process regulation, and 32 genes associated with other processes related to cold tolerance were annotated (Table 4).

## Expression level verification

We selected 14 candidate genes with potential roles in cold tolerance, including 11 annotated genes described in Table 4 and Fig. 6, and three genes with high differential expression in the common DEG set but without functional annotation. Fourteen genes encoding staphylococcal nuclease domain-containing protein 1 (TRINITY_DN22243_c0_g1), ATP synthase subunit alpha (TRINITY_DN18757_c0_g2), protein phosphatase 1A (TRINITY_DN26187_c0_g1), fructose-1,6-bisphosphatase 1 (TRINITY_DN21780_c1_g1), glyceraldehyde-3-phosphate dehydrogenase 2 (TRIN-ITY_DN22610_c0_g2), putative glutamate synthase (TRINITY_DN27207_c0_g1), cathepsin L1 (TRINITY_DN18927_c0_g1), probable low-specificity L-threonine aldolase 2 (TRINITY_DN22377_c1_g1), serine/threonine-protein kinase (TRIN-ITY_DN26090_c0_g1), fatty acid desaturase (TRINITY_DN19691_c0_g1), E3 ubiquitin-protein ligase (TRINITY_DN25425_c0_g4) and three genes annotated to unknown or uncharacterized proteins (TRINITY_DN27537_c0_g1, TRINITY_DN20229_c0_g1, and TRINITY_DN22439_c0_g5) were evaluated by qPCR (Figs. 7 and 8).

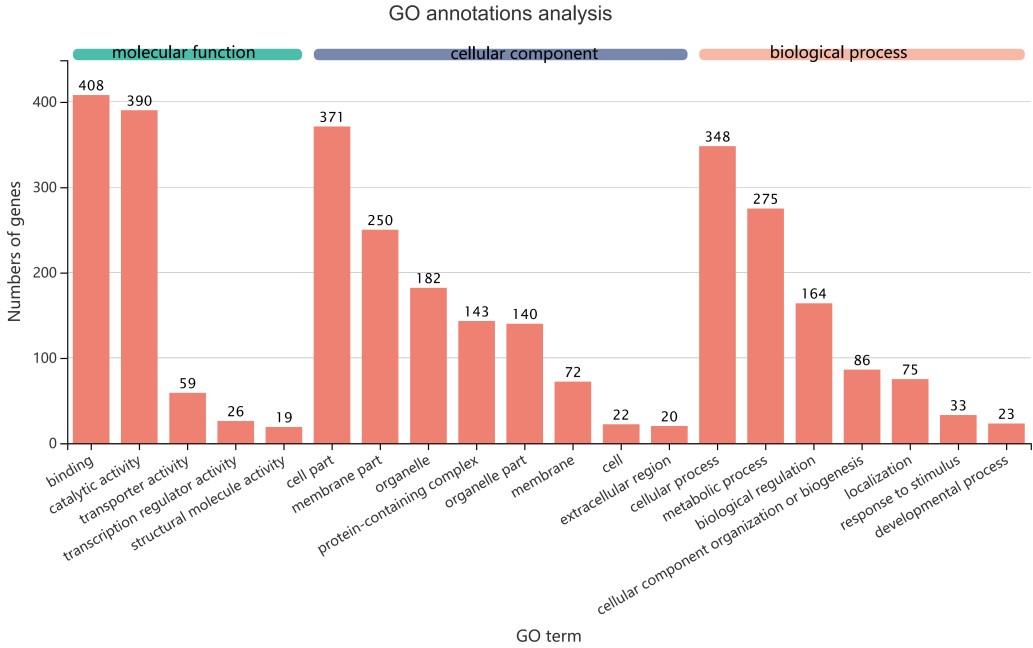

**Figure 5  GO term assignments for common DEGs between collection times in *D. valens* larvae and adults.** The ordinate indicates the number of unigenes in the secondary classification. The abscissa indicates the secondary GO classification and the three colors represent the three major branches of GO (i.e., BP, CC, and MF).

Based on the qPCR results, the expression differences in the 14 DEGs were consistent with those detected by RNA-seq, indicating that the sequencing data were accurate and reliable and can be used as a reference for subsequent research. Slight differences in fold change values may be due to operational errors and the different principles of the analysis methods for transcriptome sequencing and qPCR.

## DISCUSSION

### Transcriptome sequencing quality and analysis of differentially expressed genes

Based on searches against the Nr protein database of NCBI, 27,047 genes (45.88%) were homologous to known sequences, indicating that the transcriptome includes a large number of new genes with unknown functions. These genes may be related to the unique biological characteristics and environmental adaptability of the species. Taking May data as a control, comparing January data with May data, the result showed the 1,140 common DEGs between sampling dates in both larvae and adults, 370 were up-regulated in both adults and larvae and 505 were down-regulated, similar to the results of transcriptome analyses of the Chinese white wax scale insect (*Ericerus pela*) and the desert tenebrionid beetle (*Microdera punctipennis*), although the frequency of upregulated genes is low, these results suggest that some physiological activities remain active or are initiated in but no longer in May (*Yu et al., 2016*; *Tusong et al., 2017*).

**Table 2** KEGG pathway analysis of common DEGs between sampling dates in *D. valens* larvae and adults.

| Pathway ID | Second Category | Description | Count |
| --- | --- | --- | --- |
| map04714 | Environmental adaptation | Thermogenesis | 20 |
| map04142 | Transport and catabolism | Lysosome | 19 |
| map00230 | Nucleotide metabolism | Purine metabolism | 18 |
| map04140 | Transport and catabolism | Autophagy - animal | 18 |
| map04024 | Signal transduction | cAMP signaling pathway | 15 |
| map00190 | Energy metabolism | Oxidative phosphorylation | 14 |
| map04530 | Cellular community - eukaryotes | Tight junction | 13 |
| map00040 | Carbohydrate metabolism | Pentose and glucuronate interconversions | 12 |
| map04141 | Folding, sorting and degradation | Protein processing in endoplasmic reticulum | 12 |
| map04210 | Cell growth and death | Apoptosis | 12 |
| map04910 | Endocrine system | Insulin signaling pathway | 12 |
| map00520 | Carbohydrate metabolism | Amino sugar and nucleotide sugar metabolism | 11 |
| map04022 | Signal transduction | cGMP-PKG signaling pathway | 11 |
| map04260 | Circulatory system | Cardiac muscle contraction | 11 |
| map00240 | Nucleotide metabolism | Pyrimidine metabolism | 10 |
| map04010 | Signal transduction | MAPK signaling pathway | 10 |
| map04145 | Transport and catabolism | Phagosome | 10 |
| map04921 | Endocrine system | Oxytocin signaling pathway | 10 |
| map04723 | Nervous system | Retrograde endocannabinoid signaling | 10 |

The 370 up-regulated genes in both adult and larvae encoded several proteins that may be related to cold tolerance, including protein phosphatase, elongation of very long chain fatty acids protein, E3 SUMO-protein ligase, cytochrome P450, and putative leucine-rich repeat-containing protein. Studies show that under low-temperature and freeze-thaw conditions, protein phosphorylation and dephosphorylation are important regulatory mechanisms related to many metabolic functions. Protein kinases and phosphatases can regulate the activity of many transcription factors and participate in the differential expression of genes involved in cold and freeze resistance (*Pfister & Storey, 2006*). Overwintering insects need to neutralize or prevent the production of harmful metabolites. To modify cytotoxic metabolites, cytochrome P450 in the Antarctic midge (*Belgica antarctica*) is up-regulated during recovery from dehydration (*Benoit et al., 2009*). We speculate that cytochrome P450 plays a similar role in cold tolerance. In addition, elongation of very long chain fatty acids protein and leucine-rich repeat-containing protein contribute to the synthesis and metabolism of low-molecular-weight cryoprotectants, and other genes may be important for the maintenance of dormancy and related physiological processes.

We identified 1,140 unigenes with differential expression in both adults and larvae, accounting for only 18.72% of the DGEs in adults and 25.99% of the DGEs in larvae. Most of the DGEs were not expressed in adults and larvae but were involved in the synthesis and expression of adaptation-related genes at low temperatures, such as some heat shock proteins, the sugar transporter SWEET1, and calcium channel proteins. For cell survival, heat shock proteins are important for survival during the winter. Many

**Table 3 GO enrichment of common DEGs between sampling dates in *D. valens* larvae and adults.**

| GO category | Term type | GO code | Number | P-value |
|---|---|---|---|---|
| Carbohydrate metabolic process | BP | GO:0005975 | 65 | 2.63E−06 |
| External encapsulating structure organization | BP | GO:0045229 | 7 | 3.70E−04 |
| Cell wall organization or biogenesis | BP | GO:0071554 | 7 | 3.70E−04 |
| Cell wall organization | BP | GO:0071555 | 7 | 3.70E−04 |
| Membrane lipid metabolic process | BP | GO:0006643 | 13 | 3.76E−04 |
| Cell wall modification | BP | GO:0042545 | 4 | 1.05E−03 |
| RNA polymerase II transcriptional preinitiation complex assembly | BP | GO:0051123 | 4 | 1.05E−03 |
| Polysaccharide catabolic process | BP | GO:0000272 | 6 | 1.47E−03 |
| Organic acid biosynthetic process | BP | GO:0016053 | 18 | 1.50E−03 |
| Carboxylic acid biosynthetic process | BP | GO:0046394 | 18 | 1.50E−03 |
| Polysaccharide metabolic process | BP | GO:0005976 | 8 | 1.62E−03 |
| Small molecule metabolic process | BP | GO:0044281 | 76 | 2.13E−03 |
| Hydrolase activity, acting on glycosyl bonds | MF | GO:0016798 | 43 | 5.80E−07 |
| Hydrolase activity, hydrolyzing O-glycosyl compounds | MF | GO:0004553 | 40 | 1.06E−06 |
| Phosphatase activity | MF | GO:0016791 | 24 | 1.01E−04 |
| Hydrolase activity | MF | GO:0016787 | 213 | 1.09E−04 |
| Cellulase activity | MF | GO:0008810 | 6 | 1.65E−04 |
| Carbohydrate binding | MF | GO:0030246 | 16 | 2.26E−04 |
| Catalytic activity | MF | GO:0003824 | 396 | 2.87E−04 |
| Magnesium ion binding | MF | GO:0000287 | 13 | 8.87E−04 |
| Pectinesterase activity | MF | GO:0030599 | 4 | 1.05E−03 |
| Aspartyl esterase activity | MF | GO:0045330 | 4 | 1.05E−03 |
| Phosphatidylinositol binding | MF | GO:0035091 | 12 | 1.30E−03 |
| Phosphoric ester hydrolase activity | MF | GO:0042578 | 26 | 1.47E−03 |
| Serine-type exopeptidase activity | MF | GO:0070008 | 4 | 1.67E−03 |

insects alleviate stress by up-regulating HSP (*Storey & Storey, 2011*; *King & MacRae, 2015*; *Toxopeus, Koštál & Sinclair, 2019*; *Li, Wang & Jiang, 2019*), but only a subset of HSP genes are up-regulated, consistent with previous results (*Zhang et al., 2011*). In this study, HSP genes were up-regulated in both adult and larvae, including *hsp20*, *hsp70*, and *hsp90*. The differentially co-expressed gene set only one *hsp70* and one *hsp68*. Although the specific function of *hsp70* in survival at low temperatures in insects is not clear, it improves the response to stress by re-folding damaged proteins and re-dissolving insoluble proteins (*Craig et al., 1985*) as well as marking irreparable proteins for degradation (*Terlecky et al., 1992*). The sugar transporter SWEET1 is a cold tolerance gene based on a transcriptome analysis of the Asian lady beetle (*Harmonia axyridis*) at normal and low temperatures (*Tang et al., 2017*). In larvae, calcium channel proteins are significantly up-regulated compared to levels in adults, and calcium signaling pathways are also important in cold stress signaling (*Reddy et al., 2011*; *Denlinger et al., 2013*; *Wang et al., 2013*).

Most of Duman's work (as well as Koštál's) has centered on freeze avoiding species that deeply supercool—hence, the interest in antifreeze proteins by these authors. For lots of

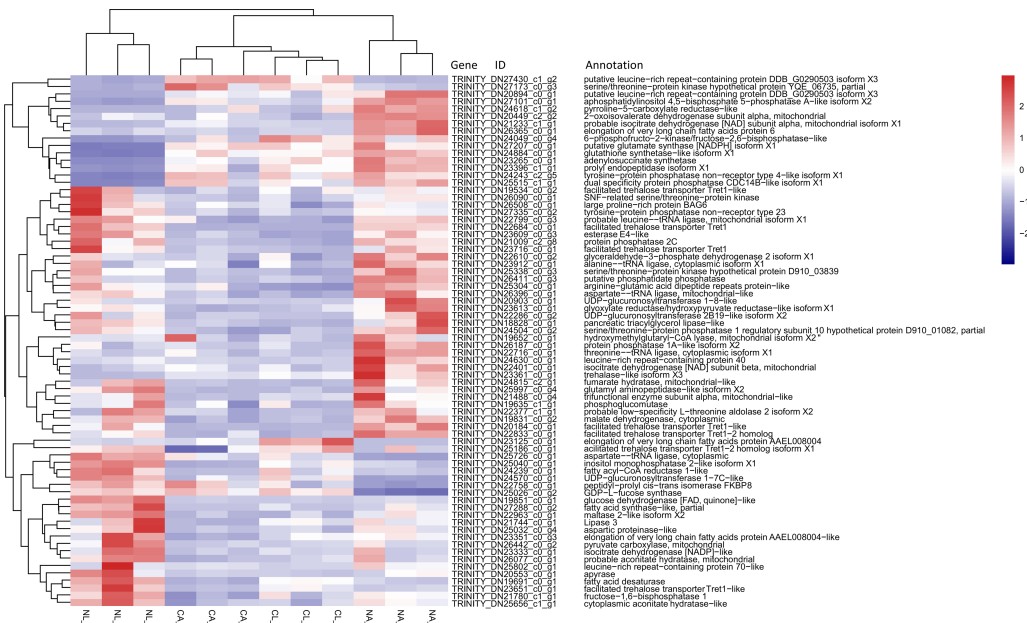

**Figure 6** **Heatmap of normalized FPKM values for DEGs related to cryoprotectant activity.** The Z-score represents the number of standard deviations from the mean. Firebrick indicates up-regulated expression, whereas navy indicates down-regulated expression. FPKM: fragments per kilobase of transcript per million fragments mapped; CL: Larva in January; NL: Larva in May; CA: Adult in January; NA: Adult in May.

other species, however, freeze tolerance is the adaptation used (*Koštál et al., 2011a*; *Koštál et al., 2011b*; *Duman, 2015*). However, anti-freeze proteins (AFPs) were not included in our significant DEG sets. It may be due to *D. valens* is a freezing-tolerant species by using the freeze tolerance strategy for subzero survival. It can permit extracellular ice formation within tissues and actively form ice-nucleating proteins (*Zhao, Yang & Gregoire, 2009*). It is possible that the relationship between transcript and protein levels is complex, and transcriptome and proteome analyses often show weak correlations between transcript and protein level (*Cui et al., 2017*).

## Analysis of genes related to cold tolerance among common DEGs

A large number of DEGs were related to the synthesis and metabolism of small cryoprotectants. Freezing-tolerant insects accumulate low-molecular-weight cryoprotectants, such as polyols (e.g., glycerol), sugars (e.g., trehalose), or amino acids (e.g., proline), to confer freezing tolerance. Elevated proline concentrations in the body can improve the survival of cold-sensitive *D. melanogaster* larvae at low temperatures, and trehalase plays a role in regulating the cold tolerance in *H.axyridis* and New Zealand alpine insects (*Imek et al., 2012*; *Wharton, 2011*; *Tang et al., 2017*). In our results, there are seven facilitated trehalose transporter Tret1 genes differentially expressed (Fig. 6). The correlation between cold tolerance and upregulation of trehalose transporters has been demonstrated in the cold-acclimated spring field cricket (*Gryllus veletis*) transcriptome. It is speculated that the *Tret-1* promotes trehalose export from the fat body into the hemolymph during

**Table 4 Common differentially expressed gene set may be related to cold tolerance.**

| Gene description | DEG number | Larval expression (number) | Adult expression (number) |
|---|---|---|---|
| Protein folding/chaperone | | | |
| HSP70 | 1 | up (1) | up(1) |
| HSP68 | 1 | down(1) | up(1) |
| Cytoskeleton | | | |
| Tubulin | 3 | down (3) | down (2); up (1) |
| Actin | 6 | down (3); up (3) | down (6) |
| Myosin | 3 | down (2); up (1) | down (3) |
| Repair/detoxicant/antioxidant | | | |
| DNA mismatch repair protein Mlh1 | 1 | up (1) | up (1) |
| Thioredoxin | 1 | up (1) | down (1) |
| Cytochrome P450 | 7 | down (3); up (4) | down (2); up (5) |
| Signal transduction | | | |
| Guanine nucleotide-binding protein | 3 | down (2); up (1) | down (2); up (1) |
| Rho guanine nucleotide exchange factor | 2 | down (1); up (1) | down (2) |
| Calcium channel protein | 3 | up (3) | down (3) |
| G kinase-anchoring protein | 1 | up (1) | up (1) |
| G-protein coupled receptor Mth2 | 1 | up (1) | up (1) |
| Cellular processes | | | |
| RNA polymerases I, II, and III subunit RPABC2 | 1 | up (1) | up (1) |
| Other genes | | | |
| Juvenile hormone epoxide hydrolase 1 | 1 | down (1) | down (1) |
| ATP synthase subunit alpha | 1 | down (1) | down (1) |
| Endonuclease-reverse transcriptase | 1 | down (1) | down (1) |
| E3 SUMO-protein ligase | 1 | up (1) | up (1) |
| E3 ubiquitin-protein ligase | 5 | down (3); up (2) | down (4); up (1) |
| Cathepsin | 6 | down (6) | down (6) |
| Zinc finger protein | 12 | down (2); up (10) | down (1); up (11) |
| NADH dehydrogenase | 2 | down (1); up (1) | down (2) |
| Elongation factor | 1 | down (1) | down (1) |
| Ribosomal protein | 2 | down (2); up (1) | down (1); up (1) |
| Staphylococcal nuclease domain-containing protein 1 | 1 | down (1) | down (1) |

**Notes.**

The samples were collected in January 2019 and May 2019, and May represents the control group. Up and down mean that the gene expression is up-regulated or down-regulated in January compared with May.

acclimation, resulting in hemolymph trehalose accumulation (*Toxopeus, Des Marteaux & Sinclair, 2019*). Pyrroline-5-carboxylate reductase is very important in the synthesis of proline. Studies have shown that *D. melanogaster*, the Antarctic midge (*B. antarctica*) and New Zealand stick insects up-regulate the pyrroline-5-carboxylate reductase in association with proline accumulation (*MacMillan et al., 2017*; *Teets et al., 2012a*; *Teets et al., 2012b*; *Dennis et al., 2015*). Fatty acid synthase-related genes are differentially expressed at low temperatures, it has been showned that the production of fatty acids maintained the

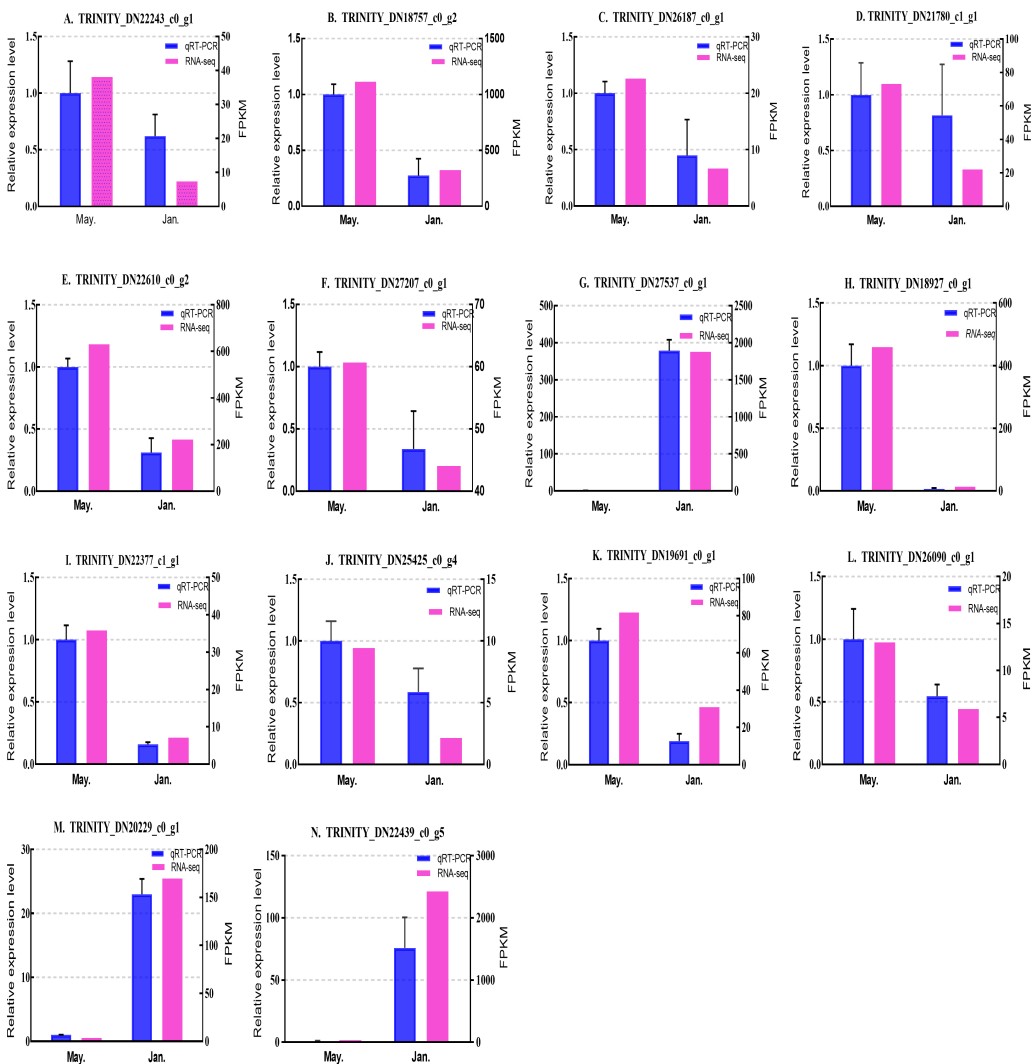

**Figure 7** **Validation expression patterns in *D. valens* adults by qPCR.** The samples were collected in January 2019 and May 2019. (A) Staphylococcal nuclease domain-containing protein 1; (B) ATP synthase subunit alpha; (C) protein phosphatase 1A; (D) fructose-1,6-bisphosphatase 1; (E) glyceraldehyde-3-phosphate dehydrogenase 2; (F) putative glutamate synthase; (G) unknown; (H) cathepsin L1; (I) probable low-specificity L-threonine aldolase 2; (J) E3 ubiquitin-protein ligase; (K) fatty acid desaturase; (L) serine/threonine-protein kinase; (M) uncharacterized protein; (N) uncharacterized protein. The abscissa indicates different collection dates. The left ordinate represents the qPCR-based expression levels and the right ordinate represents the RNA-seq-based expression levels. The error bar refers to SEM and the N value represents three repetitions. FPKM: fragments per kilobase of transcript per million fragments mapped.

homoeoviscosity of the cellular membranes, which contributes to enhanced cold tolerance (*Kayukawa et al., 2007*; *Michaud & Denlinger, 2007*; *Goto et al., 2010*).

A number of genes we observed in *D. valens* have been suggested as contributing to the cold tolerance of insects including cathepsin (*Dennis et al., 2015*), fructose-2,6-bisphosphatase (*Fraser et al., 2017*), pancreatic triacylglycerol lipase (*Bonnett et al., 2012*;

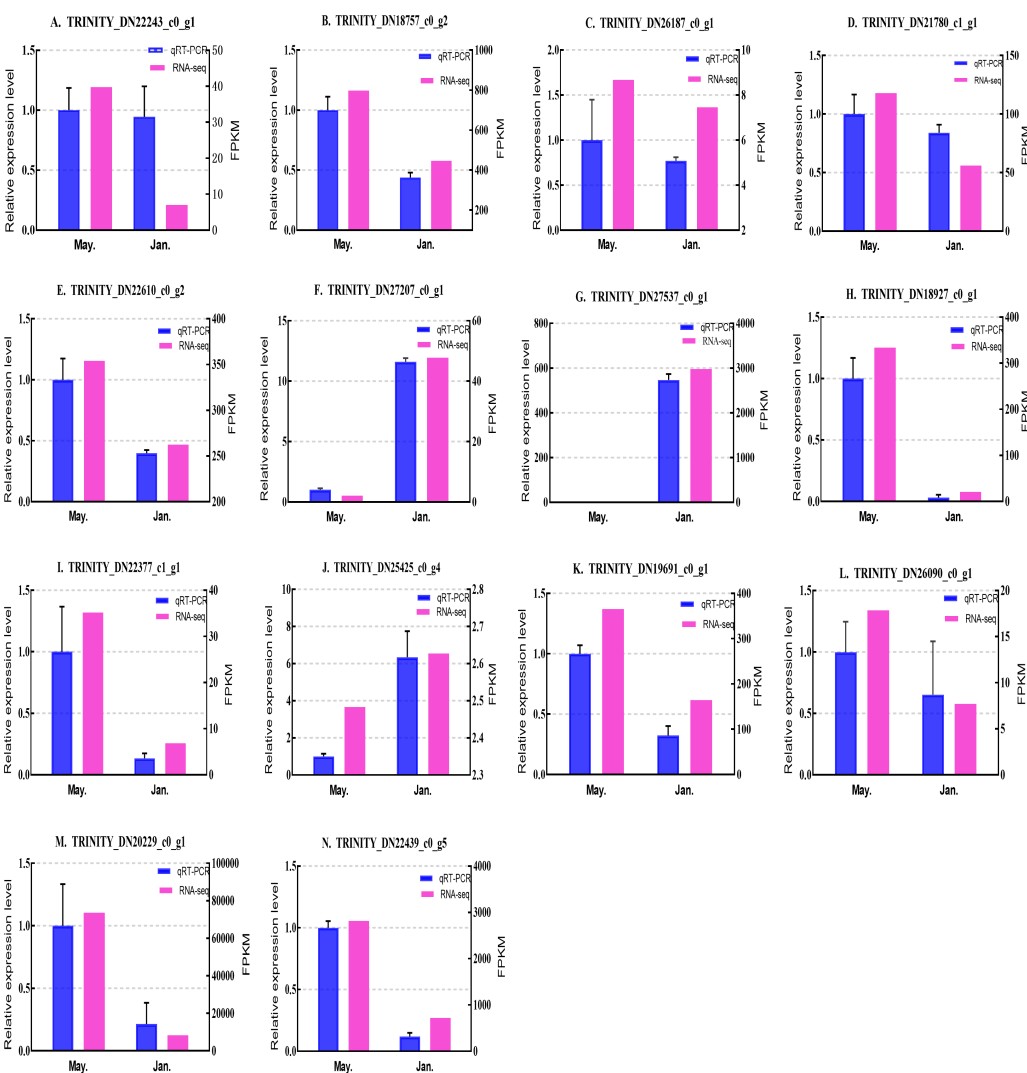

**Figure 8  Validation expression patterns in *D. valens* larvae by qPCR.** The samples were collected in January 2019 and May 2019. (A) Staphylococcal nuclease domain-containing protein 1; (B) ATP synthase subunit alpha; (C) protein phosphatase 1A; (D) fructose-1,6-bisphosphatase 1; (E) glyceraldehyde-3-phosphate dehydrogenase 2; (F) putative glutamate synthase; (G) unknown; (H) cathepsin L1; (I) probable low-specificity L-threonine aldolase 2; (J) E3 ubiquitin-protein ligase; (K) fatty acid desaturase; (L) serine/threonine-protein kinase; (M) uncharacterized protein; (N) uncharacterized protein. The abscissa indicates different collection dates. The left ordinate represents the qPCR-based expression levels and the right ordinate represents the RNA-seq-based expression levels. The error bar refers to SEM and the N value represents three repetitions. FPKM: fragments per kilobase of transcript per million fragments mapped.

*Fraser et al., 2017*), zinc finger proteins (*Xu et al., 2017*), glyceraldehyde-3-phosphate dehydrogenase 2 (*Fraser et al., 2017*), NADH dehydrogenase (*Dunning et al., 2013*), ATP synthase subunit alpha (*Li & Denlinger, 2008*), elongation factor (*Tusong et al., 2017*), staphylococcal nuclease domain-containing protein 1 (*Dunning et al., 2013*), etc. We

hypothesize that changes in the expression of these genes increase the cold tolerance of *D. valens*.

In this study, 12 cytoskeletal genes were differentially expressed (e.g., *Tubulin*, *Actin*, and *Myosin*; Table 4), and cytoskeletal genes have been identified to play an important role in adaptation to low temperature in many other insects such as the northern house mosquito (*Culex pipiens*) (*Kim et al., 2006*), the drosophilid fly (*Chymomyza costata*) (*Stetina et al., 2018*), *M. punctipennis* (*Tusong et al., 2017*), *D. melanogaster* (*Dennis et al., 2016*), the fall field cricket (*Gryllus pennsylvanicus*) (*Des Marteaux et al., 2017*). We speculate that the differential expression of cytoskeletal genes are necessary particularly for maintaining cell integrity during the wintering period. In addition, studies have shown that the drosophilid fly (*C. costata*) (*Poupardin et al., 2015*), the spring field cricket (*G. vele tis*) (*Toxopeus, Des Marteaux & Sinclair, 2019*) and the flesh fly (*Sarcophaga bullata*) (*Teets et al., 2012a*; *Teets et al., 2012b*) upregulate some signaling genes after cold treatment. The expression of some signal transduction-related genes in *D. valens* (such as guanine nucleotide-binding protein, $Ca^{2+}$ channel, G kinase-anchoring protein, G-protein coupled receptor Mth2; Table 4) were altered with temperature, but it is difficult to predict how these changes enhance cold cold tolerance.

In addition, low-molecular-weight cryoprotectants are related to anabolism genes, cytoskeleton protein genes, and repair-related genes (*Toxopeus & Sinclair, 2018*), in addition to many unannotated genes. Although the specific mechanism of action of these genes in insects is unclear, they may also play an important role in cold resistance. Cold tolerance in insects is a complex process, involving interactions among multiple genes (*Toxopeus & Sinclair, 2018*).

DEGs involved in cold tolerance are expected to be up-regulated under cold stress; however, most DEGs in our study were down-regulated. One is that the metabolic rate of *D. valens* in January may be much lower than in May. Studies have shown that when spring becomes warmer, almost all enzymes involved in carbohydrate metabolism (glycolysis, citric acid cycle, glycerol biosynthesis), oxidative stress tolerance, detoxification mechanism, cryoprotectant metabolism protein levels decrease than winter, the strong winter metabolic suppression inhibition can improve the supercooling ability, which has been verified in the overwintering mountain pine beetles (*Lester & Irwin, 2012*; *Bonnett et al., 2012*; *Robert et al., 2016*). Second, it is possible that genes and proteins related to low temperature adaptation were upregulated at the beginning of the winter and that expression dropped when they were no longer needed, with the saturation of expression levels in late January (*Fraser et al., 2017*). Therefore, protein expression levels should be a focus of future research.

## Cold tolerance-associated GO and KEGG enrichment analysis

GO terms that were significantly enriched in the down-regulated gene set were carbohydrate metabolism processes, hydrolase activity, cellulase activity, catalytic activity and so on. It is similar to the GO enrichment results of the differentially expressed genes of the European spurge hawkmoth (*Hyles euphorbiae*) under cold treatment (*Barth et al., 2018*). Under stress, metabolic processes are down-regulated in many organisms, and

the down-regulation of hydrolase and cellulase activity may be the result of metabolic down-regulation. The downregulation of carbohydrate metabolism suggests that the relationship between energy utilization and temperature is altered in insects during the winter by changes in thermal sensitivity and the inhibition of metabolic rates; these insects adapt to low temperature by changing energy metabolism (*Storey & Storey, 1990*; *Sinclair, 2015*).

In the co-expressed gene set, a KEGG enrichment analysis showed enrichment for pentose and glucuronate interconversions, lysosome, autophagy—animal, and so on. The autophagy pathway shows that *D. valens* activates its own protective mechanism to repair and clear damaged cells under stress to maintain body operation. We hypothesize that the ability of *D. valens* to clear cold damaged cells may increase in winter (*Allen & Baehrecke, 2020*).

The pathways basal transcription factors and homologous recombination were significantly enriched in the up-regulated gene set. Studies have shown that RNA polymerase II-dependent transcription involves the regulation of basic transcription factors, and RNA polymerases I, II, and III are also up-regulated in the transcripts of cold-acclimated *G. veletis* (*Toxopeus, Des Marteaux & Sinclair, 2019*). Some significantly enriched pathways (pentose and glucuronate interconversions, galactose metabolism, fatty acid elongation, citrate cycle (TCA cycle), lysosome, and autophagy—animal) were detected for common down-regulated DEGs. Similar results were obtained in the transcriptome response to cold stress of the ladybird (*Cryptolaemus montrouzieri*) (*Zhang et al., 2015*) and *Galeruca daurica* (Joannis) (*Zhou et al., 2019*). These metabolic pathways indicated that overwintering *D. valens* may produce or consume less energy, consistent with the general response of winter insects to low temperatures by inhibiting the metabolic rate (*Sinclair, 2015*).

In conclusion, we used RNA-Seq technology to analyze the *D. valens* transcriptome at different periods in the field. We identified many common genes and pathways with potentially important roles in overwintering, improving our understanding of the molecular basis for survival in low temperatures. This study provides an overview of candidate genes associated with cold tolerance in insects, and further validation and functional analyses are needed. Our data will facilitate further molecular studies of cold tolerance in *D. valens* and provide new insights into insect adaptation to harsh environments.

## CONCLUSIONS

In a comparative transcriptome analysis of *D. valens* in January and May, we used the samples collected in May as controls and detected 4,387 and 6,091 DEGs in larvae and adults, respectively, including 1,140 genes that differentially expressed at both stages, among which 538 genes were up-regulated and 602 genes were down-regulated in larvae; in adults, 467 genes were up-regulated and 673 genes were down-regulated. During the overwintering process, one strategy for survival in low temperatures is the synthesis of low-molecular-weight antifreeze substances, anti-oxidative stress factors, molecular chaperones, and signal transduction factors. The DEGs were enriched for the GO terms cellulase activity, hydrolase activity, and carbohydrate metabolism and for the lysosomal

and pentose and glucuronate interconversions metabolic pathways. We identified 140 genes that may be related to cold tolerance in the common DEGs, some of which are associated with cold tolerance based on previous studies. The results provided basic data for the subsequent discovery of key genes for cold tolerance in *D. valens* and the discovery of molecular mechanisms underlying this trait.

## ACKNOWLEDGEMENTS

The data were analyzed on the free online platform of Majorbio Cloud Platform, Qingqing Liu and Ling Wang provided professional service. Field work of sample collection was partially supported by the Heilihe National Nature Reserve, Chifeng, Inner Mongolia, China.

### Funding

This work was supported by the National Natural Science Foundation of China (NO. 31870642). The funders had no role in study design, data collection and analysis, decision to publish, or preparation of the manuscript.

### Grant Disclosures

The following grant information was disclosed by the authors:
National Natural Science Foundation of China: 31870642.

### Competing Interests

The authors declare there are no competing interests.

### Author Contributions

- Dongfang Zhao conceived and designed the experiments, performed the experiments, analyzed the data, prepared figures and/or tables, authored or reviewed drafts of the paper, and approved the final draft.
- Chunchun Zheng performed the experiments, prepared figures and/or tables, and approved the final draft.
- Fengming Shi performed the experiments, prepared figures and/or tables, authored or reviewed drafts of the paper, and approved the final draft.
- Yabei Xu conceived and designed the experiments, prepared figures and/or tables, and approved the final draft.
- Shixiang Zong and Jing Tao conceived and designed the experiments, authored or reviewed drafts of the paper, and approved the final draft.

### Field Study Permissions

The following information was supplied relating to field study approvals (i.e., approving body and any reference numbers):

Field sample collection has been approved by Heihe National Nature Reserve in Inner Mongolia. Liu Yushan, head of Chifeng Forest Protection Station in Inner Mongolia, approved sample collection.

## DNA Deposition

The following information was supplied regarding the deposition of DNA sequences:

The Dendroctonus valens transcriptomes sequences are available at NCBI SRA: SRR11210693 to SRR11210704.

They are also accessible via Figshare:

Zhao, Dongfang (2021): Dendroctonus valens. figshare. Dataset. https://doi.org/10.6084/m9.figshare.11998458.v1.

Zhao, Dongfang (2021): Dendroctonus valens. figshare. Dataset. https://doi.org/10.6084/m9.figshare.12005409.v1.

Zhao, Dongfang (2021): Dendroctonus valens. figshare. Dataset. https://doi.org/10.6084/m9.figshare.12005142.v1.

## Data Availability

Data are available at NCBI SRA (PRJNA609406) and at NCBI GEO (GSE156139).

## Supplemental Information

Supplemental information for this article can be found online at http://dx.doi.org/10.7717/peerj.10864#supplemental-information.

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
