# Peer review of "Expression analysis of genes related to cold tolerance in Dendroctonus valens"

_PeerJ, doi:10.7717/peerj.10864_

## Round 0.1 · original submission · Major Revisions

The manuscript relates to a topic of great interest but it is currently missing numerous details from the methods. For example, many of the parameters required to run the programs used in this manuscript are missing. This information is vital for other readers to evaluate the manuscript and for readers to model similar analyses from in the future. In addition, the introduction (two paragraphs) is quite insufficient. The manuscript is being returned in its current form but I would recommend resubmission once these issues are resolved.

---

## Round 0.2 · Major Revisions

There are a number of minor revisions to consider in addition to five major concerns. Particular attention should be paid to the following:

1. Apparent nematode contamination in some of the data, as indicated by one of the reviewers. This should be removed and any impacts on subsequent analyses dealt with.

2. Clarity in what results are being discussed and compared. At present it is not easy to follow the trail of logic with respect to which genes/time points etc. are being compared and why.

3. Availability of data. Please ensure that the data are readily available in an appropriate repository. Currently, much of the sequence information is not found in SRA.

4. Time points. The time points at which the observations/material were collected may not be the best for answering the questions at hand. Please make sure that this potential limitation is addressed in the manuscript, or provide some more justification for why it is unlikely to be a limitation.

5. Grammar. There are numerous typos throughout the manuscript that need to be fixed. Also, some sentences are unclear due to grammatical errors.

Reviewer 1 ·

Basic reporting

The authors say "All raw sequence data were deposited in the NCBI Sequence Read Archive (SRA) database (http://www.ncbi.nlm.nih.gov/subs/) under BioProject accession no. PRJNA609406. However, I was unable to find this database on the NCBI BioProject database. Could the authors please check this. Perhaps the submission has not been fully finalized or has not yet been released to the public.

Experimental design

no comment

Validity of the findings

no comment

Additional comments

Overall, the manuscript was well written and the data are strong. Various comments are below. My greatest problem with the manuscript is that although many DEGs are identified and 112 candidate genes explored, the authors consistently fail to clearly state what they are comparing in both the Results and Discussion. We are told that genes are differentially regulated but not whether upregulation means that genes are expressed more strongly in January animals than in May animals or more strongly in May animals than in January ones (also what is the comparable definition for down-regulation). Even Table 4 does not define what the comparison is for “down” and “up”. Please make an effort to clarify sentences in the Results and Discussion so that readers can better appreciate which genes are apparently upregulated in support of winter survival vs those that are suppressed in the winter OR alternately which are not expressed in January but upregulated in the spring when eating and growth resume. This would help readers to understand the possible functionality of various differentially expressed genes with respect to cold hardiness versus spring resumption of growth.

1. It would be helpful to include the common name of this beetle in the Abstract and/or the Introduction to attract readers who do not recognize the scientific name since, for example, pine beetles are well known in North America for their widespread devastation of pine forests.
For example, the second sentence of the Abstract could say “Adults and larvae of this bark beetle winter at….”
For example: the first line of the Introduction could be “Dendroctonus valens, the red turpentine beetle, is a species of bark beetle that mainly attacks the base of the trunk….”

2. Line 65. The authors might also be interested in an early review by RW Salt: Principles of insect cold hardiness, Annual Review of Entomology, 1961, 6: 55-74.
3. Line 123. Please indicate if the biological material was “whole animal” or whether a selected part of the animal was used. Also, did each sample consist of a single individual or were multiple animals combined per sample or per biological replicate.
4. Line 125-6. Please define what CL, NL, CA and NA mean for readers. Clearly the L and A would be larvae and adults but C and N are not defined.
5. Line 305. Is “degradation” the correct word when later in the sentence the authors include “21 genes involved in the synthesis of low molecular weight polyols”. Perhaps the 44 genes are “related to cryoprotectant metabolism” including both synthesis and/or degradation.
6. Lines 338-343. Please clarify the parameters of “up-regulated” and “down-regulated”. Are you comparing the May data to the winter January situation OR the January data with the later May data. This is important for data interpretation – for example, we would expect a reduction in transcripts coding for polyol synthesis genes in May, as compared with January, but not the reverse. This information is also needed for readers to be able to interpret and understand the results reported in lines 343-347 as well as other data mentioned later
7. Lines 342-343. “…these results suggest that some physiological activities remain active or are initiated” – when? In May as compared with earlier in January? Or were active in January but no longer in May?

8. Line 362-363. Trehalose is NOT a protein, it’s the normal blood sugar of insects but is often elevated as part of the cryoprotectant system in winter. Do you mean “trehalase”, the enzyme that cleaves the disaccharide into 2 glucose molecules?
9. Lines 388-395. Can the authors tell readers whether D. valens uses the freeze-avoiding or the freeze tolerance strategy for subzero survival? Freeze-avoiding insects that deeply supercool will have antifreeze proteins whereas those that are freeze tolerant are much less likely to have such proteins and instead have ice-nucleating proteins that help the insect to regulate ice formation in specific body compartments. If this has not been tested for D. valens, then perhaps this has been tested for D. ponderosae. Most beetles use the freeze avoidance strategy but it would be best to indicate what is known for D. valens or for pine beetles in general
10. Line 388 is better stated as “Studies show that cold tolerance in many insect species is directly related to antifreeze proteins…”. Most of Duman’s work (as well as Kostal’s) has centered on freeze avoiding species that deeply supercool – hence, the interest in antifreeze proteins by these authors. For lots of other species, however, freeze tolerance is the adaptation used.
11. Line 404-405 needs a reference citation for the Galeruca daurica statement.
12. Line 444 and following. In these conclusions, please make it clear whether the enriched DEGs and the 112 screened genes were ones that were higher in January (ie. when cold hardiness is needed) or higher in May.
13. For Table 4, please define what up and down are. Does UP mean the gene in was higher in January or higher in May? Same question for DOWN. Please define this in the legend of the table. This is crucial for readers to be able to understand the data. Also, I would suggest adding 2019 to the Table legend as in “collected in January 2019 or in May 2019” so that readers realize that the sampling timeline was January first and then May later.
14. For the legends of Fig. 7 and 8, please define what the error bars are (SD or SEM) as well as the value for N (number of replicates). It would also be useful to restate that animals were sampled in January 2019 and then May 2019.

Reviewer 2 ·

Basic reporting

This manuscript is generally clear, well-written, and well-organized. However, throughout spaces are sometimes missing between words, or when a word is followed by a parenthesis. Sometimes an extra space appears between the end of a word and a punctuation mark.

As described below in Comment to Authors:
-There are references lacking for some of the software used.
-The authors do not provide the relevant context to their work by omitting prior relevant research in other Dendroctonus beetles.
-Public deposit of transcriptome assembly and gene expression analyses should be done (NCBI TSA and GEO, respectively).

Experimental design

Although the experimental design is appropriate, I am not sure that the two timepoints chosen best reflect the changes that are happening to develop freezing resistance during the winter, as the winter timepoint would be after the insects are likely to have obtained their maximum freezing resistance and would have a reduced metabolic rate. As the authors suggest, perhaps proteomic (and metabolite) analyses would better answer what is happening at the January timepoints.
See General comments for the author

Validity of the findings

Authors should cite and incorporate their results with those previously reported in other Dendroctonus.
See General comments for the author

Additional comments

DNA data checks
Have you checked the authors data deposition statement?
Link does not exist
Can you access the deposited data? No
Has the data been deposited correctly? No
Is the deposition information noted in the manuscript?
Authors indicate that the raw reads have been submitted to NCBI SRA, but not the assembly to NCBI TSA, nor the gene expression results to NCBI GEO. These data should also be included as part of a complete set of data deposition of RNAseq transcriptomic data.
Field study
Have you checked the authors field study permits?
Link does not exist
Are the field study permits appropriate?
Cannot determine this

Comments:
Lines 1 and 451: I don’t think “screening” is appropriate here. It implies that they were assayed for function. For the title, remove “Screening and”. For line 451, change to “identified”.
Line 116: What instar were the larvae? How was this determined?
Line 118, modify as indicated for clarity: (Longitude: 118.43° E; Latitude: 41.41° N; Altitude: 1050 m)
Lines 118-119: How were these temperatures obtained?
Lines 133-134: What type of library was constructed? Manufacturer? Single or paired-end, stranded or un-stranded? What length of reads were generated?
Line 144: why “reassembled”? Was it previously assembled?
Line 175: Colin isn’t the last name of this author.
Line 208, modify as indicated for clarity: “…melting curves. Each reaction was performed for three biological and three technical replicates.”
Line 198: TUB is used as one of the reference genes (for larvae), yet in Table 4, it shows that two tubulin transcripts are differentially regulated in larvae. The transcript ID of TUB (and PRS) should be indicated, and there should be an explanation of which tubulin is used as TUB vs. those shown as DEG in Table 4.
Figure S1 needs a description included
Table S1: Amplicon sizes are lacking for TUB and PRS18. Primer-pair efficiencies are not shown at all nor mentioned in the M&M.
Line 150: What criteria was used for clustering with CD-HIT?
Lines 152 and 219 and Table S2: BUSCO information is not properly presented. What dataset was used? What does BUSCO score of 96.3%(16.7%) mean? Use the recommended format described here: https://busco.ezlab.org/busco_userguide.html#reporting-busco
Line 163: reference for DIAMOND?
Line 165: reference for BLAST2GO?
Line 167: reference for KOBAS?
Line 187: reference for goatools?
Line 199: reference for Primer 3Plus?
Line 209, modify as indicated for clarity: “…experimental data were normalized…”
Line 217, modify as indicated for clarity: “…After clustering and reducing redundancy…”
Line 238: This sentence is confusing because you mention FPKMs but DESeq2 uses counts for DEG analysis.
Lines 278-280: Cold tolerance is a process that likely begins in the fall with decreasing temperatures. By January, the beetles should already be well-adapted to the winter cold and their metabolic rate would be quite low, and thus I question whether the beetles are RAPIDLY synthesizing small molecules, such as sugars, as antifreeze protectants at this time.
Lines 291-293: Is there any information about this in other bark beetles?
Line 352: “…cytotoxic P450 enzymes…” implies that these enzymes are cytotoxic, when I think you really mean that these enzymes modify cytotoxic metabolites. Reword.
Line 353: “…cytochrome P450…” singular or plural?
Lines 345 and 354: “very long-chain fatty acids” are not proteins/genes. Do you mean proteins/genes in fatty acid biosynthesis?
Line 361: “…freezing tolerance.” Freezing tolerance is a completely different adaptive strategy to cold temperatures than freezing-resistance (avoidance). What information is available in this and other related bark beetles on these two mechanisms? Please be sure to use the correct terminology throughout.
Line 396: The metabolic rate of bark beetles in January at -12°C is probably going to be much lower than in July. Thus, could the abundance of down-regulated genes in winter insect merely reflect this decrease in metabolic rate, rather than genes inherently important to cold tolerance?
Lines 397-399: Maybe this has already been explored in another bark beetle:
Bonnett, T.R., Robert, J.A., Pitt, C., Fraser, J.D., Keeling, C.I., Bohlmann, J., and Huber, D.P. 2012. Global and comparative proteomic profiling of overwintering and developing mountain pine beetle, Dendroctonus ponderosae (Coleoptera: Curculionidae), larvae. Insect Biochemistry and Molecular Biology 42: 890-901. doi:10.1016/j.ibmb.2012.08.003.
Fraser, J.D., Bonnett, T.R., Keeling, C.I., and Huber, D.P.W. 2017. Seasonal shifts in accumulation of glycerol biosynthetic gene transcripts in mountain pine beetle, Dendroctonus ponderosae Hopkins (Coleoptera: Curculionidae), larvae. PeerJ 5: e3284. doi:10.7717/peerj.3284.
Robert, J.A. et al. 2016. Gene expression analysis of overwintering mountain pine beetle larvae suggests multiple systems involved in overwintering stress, cold hardiness, and preparation for spring development. PeerJ 4: e2109. doi:10.7717/peerj.2109.
Line 438: -18.7°C Where does this temperature value come from?
Fig. 2C: Most of the non-Dendroctonus species are nematode, not insects at all. This suggests that there is significant contamination in the RNAseq data (>25%), most likely from nematodes in/on the beetles. It would seem that you already have the information from the BLAST and DIAMOND results to properly remove these sequences from your assembly. These need to be removed from the assembly prior to DEG analysis. Transcripts could be incorrectly identified as differentially expressed merely because they are nematode transcripts, and perhaps only adults at a certain time carry these nematodes. You also loose statistical power for identifying differences in real beetle transcripts because of these spurious transcripts. You will find that NCBI will also flag these transcripts when you submit this assembly to NCBI TSA.
Throughout, spaces are sometimes missing between words, or when a word is followed by a parenthesis. Sometimes an extra space appears between the end of a word and a punctuation mark.

---

## Round 0.3 · Minor Revisions

The manuscript has been improved and addresses the majority of the reviewer concerns. However, the outstanding issues regarding the use of FPKMs with DESeq2, and the NCBI submission of the transcriptome, really need to be better addressed. PeerJ emphasizes appropriate and rigorous methodological approaches. As such, it is important to resolve, or at least provide better justification for why these two issues remain.

Reviewer 2 ·

Basic reporting

The revised manuscript has addressed most of the prior concerns by the reviewers.

Experimental design

The revised manuscript and the rebuttal has addressed most of the prior concerns by the reviewers.

Validity of the findings

The validity of the findings may have been negatively affected by importing FPKMs into DESeq2, which it specifically warns against.

Additional comments

The revised manuscript has addressed most of the prior concerns by the reviewers. I identify the following:

Lines 62-63: This sentence doesn't have support. I would suggest deleting it. Freezing-tolerant and freezing-avoidant insects are not novel, so the mechanism is not as yet described, novel.

Line 128: mature -> late instar

Line 131: Inner Mongolia

Line 141: animal -> insect
Line 141: "Each sample consisted of..."
Line 141: repeats -> replicates

Lines 184 and 253, Table S2: There is no invertebrate dataset for V3 of BUSCO (https://busco-archive.ezlab.org/v3/frame_wget.html). What database and what version obd9(?) was used.
as per BUSCO's website, Reporting BUSCO:
Report results in simple BUSCO notation: C:89.0%[S:85.8%,D:3.2%],F:6.9%,M:4.1%,n:3023
Report the BUSCO set(s) you used for your assessments. Mention the creation date of the dataset, not only the name, e.g. archaea_odb10 (2019-01-04).
Report the BUSCO options you used.
Only the first 2 or 3 values are shown. Show them all as recommended, C:89.0%[S:85.8%,D:3.2%],F:6.9%,M:4.1%,n:3023.

Line 207: DESeq2 documentation specifically states that FPKMs should not be used as input to DESeq2, as mentioned in review of version 1 of this manuscript. Submitting normalized FPKMs will over-normalize your data and will likely negatively affect the statistical analysis of your data. The data from RSEM can be correctly imported into DESeq2 with the tximport package, which it seems you haven't used if you are incorrectly importing FPKMs into DESeq2.

Lines 264-5: "The sample" is confusing. Was only one sample contaminated? Do you mean that some sequences were contaminated? Was a sample deleted, or sequences deleted?

Line 403: accents missing from Koštál.

Other: It is concerning that the transcriptome is not acceptable to NCBI TSA due to errors. You do not specify why, so we cannot assess whether this is something that could affect the conclusions drawn, such as the prior nematode contamination you have now removed. Adapter or other contamination? The Sequin file included with the manuscript is not an appropriate alternative. I see that the transcriptome contigs are included with the GEO submission. Thus, interested readers in the future will be able to access the sequences that way. However, without submission to TSA, your data will be invisible to BLAST searches etc., which diminishes the value of your data generated, missing a wider consumer of the data.

---

## Round 0.4 · Minor Revisions

I suggest a few more minor edits to correct some grammatical errors. Overall, the authors have done an excellent job of addressing the reviewer comments.

First sentence of the abstract (line 27): Please revise this sentence to "Pine beetles are well known in North America for their widespread devastation of pine forests. However, Dendroctonus valens LeConte is an important invasive forest pest in China also."

Line 53: Please include the full genus name as it is the first use of this taxonomic name.

Line 58 (and elsewhere): Please use the full genus name at the beginning of sentences.

Line 67: there is a typo - there should be a space between 'tocharacterize'

Line 103-104: Why is it worth noting that cold tolerance has been found in D. melanogaster? Either expand this information or remove it.

Line 108: First use of HSP abbreviation - please write in full.

Line 122: metabolic is spelled incorrectly

Throughout the manuscript: there are times when numbers are presented without commas and times when numbers are presented with (e.g., 10,000 vs 10000). Please pick one and make the presentation of numbers consistent.

Line 339: there are some extra spaces here

Line 373-374, line 386, line 425, line 428, line 432, line 448-450, line 453-454, line 480, line 499-500: Can you please provide the common name for these species for those readers less familiar?

Throughout the manuscript: please provide the full name of the pathways/genes being referred to at the first instance of the use of the abbreviated name (e.g. AFP, HSP).

Thank you.

---

## Round 0.5 · accepted · Accept

The authors have revised the manuscript sufficiently.